# Effects of climate and environmental heterogeneity on the phylogenetic structure of regional angiosperm floras worldwide

Hong Qian [1,2] ✉, Shenhua Qian [3,4] ✉, Jian Zhang [5,6] & Michael Kessler[7] ✉

The tendency of species to retain ancestral ecological distributions (phylogenetic niche conservatism) is thought to influence which species from a species pool can persist in a particular environment. Thus, investigating the relationships between measures of phylogenetic structure and environmental variables at a global scale can help understand the variation in species richness and phylogenetic structure in biological assemblages across the world. Here, we analyze a comprehensive data set including 341,846 species in 391 angiosperm floras worldwide to explore the relationships between measures of phylogenetic structure and environmental variables for angiosperms in regional floras across the world and for each of individual continental (biogeographic) regions. We find that the global phylogenetic structure of angiosperms shows clear and meaningful relationships with environmental factors. Current climatic variables have the highest predictive power, especially on phylogenetic metrics reflecting recent evolutionary relationships that are also related to current environmental heterogeneity, presumably because this favors plant speciation in various ways. We also find evidence that past climatic conditions, and particularly refugial conditions, play an important role in determining the phylogenetic structure of regional floras. The relationships between environmental conditions and phylogenetic metrics differ between continents, reflecting the different evolutionary histories of their floras.

The species composition of biological communities in any given geographic area is a result of the interaction of ecological, evolutionary, and biogeographic processes[1]. The extent to which each of these processes has contributed to community assembly varies among regions. Biological interactions such as competition and symbioses may play a role in community assembly at a local scale, but at a large spatial scale, evolutionary processes, dispersal, and environmental filtering are thought to play a major role in determining the species composition in an area, whereby species that cannot survive and reproduce in the physical environment of the area are excluded[2]. Because species differ in their niches, it is a truism that niches must be conserved on phylogenies, at least to some extent. In other words, the

[1]CAS Key Laboratory for Plant Diversity and Biogeography of East Asia, Kunming Institute of Botany, Chinese Academy of Sciences, Kunming 650201, China. [2]Research and Collections Center, Illinois State Museum, 1011 East Ash Street, Springfield, IL 62703, USA. [3]Key Laboratory of the Three Gorges Reservoir Region's Eco-Environment, Ministry of Education, Chongqing University, Chongqing 400045, China. [4]College of Environment and Ecology, Chongqing University, Chongqing 400045, China. [5]Center for Global Change and Complex Ecosystems, Zhejiang Tiantong Forest Ecosystem National Observation and Research Station, School of Ecological and Environmental Sciences, East China Normal University, 200241 Shanghai, China. [6]Shanghai Institute of Pollution Control and Ecological Security, Shanghai 200092, China. [7]Department of Systematic and Evolutionary Botany, University of Zurich, Zurich, Switzerland. ✉e-mail: hqian@museum.state.il.us; qian@cqu.edu.cn; michael.kessler@systbot.uzh.ch

ability of a species to persist in a particular set of ecological conditions is constrained by its evolutionary history[3,4]. Phylogenetic niche conservatism, i.e., the tendency of species to retain ancestral ecological requirements, is thought to influence which species from a regional species pool can persist in a particular environment[3].

Angiosperms (i.e., flowering plants) are the major components of most terrestrial ecosystems across the world. They diversified during the Cretaceous and Tertiary when equable conditions of warm (tropical) climate were widespread across the globe[5–7]. Temperatures decreased more quickly at higher latitudes during the global cooling initiated in the early Eocene[8,9], so that the gradient of temperature from the equator to the poles became steeper as the time approached the Pleistocene, where cycles of glaciation forced tropical lineages at higher latitudes to withdraw to lower (warmer) latitudes or to evolve tolerance for colder temperatures unless they went extinct. Because ecological traits are phylogenetically conserved[10] and an evolutionary event can rarely produce a clade that is able to survive and reproduce in novel climatic conditions such as freezing temperatures[3,4,11,12], relatively few clades have crossed the ecophysiological barrier into cold environments. Thus, those clades that have crossed it are likely to be a phylogenetically clustered subset of the global pool, and diversification within these clades has likely strengthened this initial phylogenetic clustering over time by forming groups of closely related species[13], although distantly related lineages may also evolve traits to tolerate freezing climates[14]. This leads to the prediction that for clades that originated and diversified in warm climates, such as angiosperms as a whole, their species in communities located in colder or drier climates should, on average, be more closely related to each other (more phylogenetically clustered) and phylogenetic diversity in the communities should be low[3,15]. This proposition is commonly known as the tropical niche conservatism hypothesis[3].

This hypothesis and other related processes are reflected in the phylogenetic composition of biological communities. As detailed above, clades are expected to be most phylogenetically diverse under conditions similar to those where they originated and become phylogenetically more selective (i.e., clustered) as different ecological conditions are approached to which only a few clades have become adapted[3]. This general pattern can further be refined by focusing on different aspects of phylogeny, e.g., by employing metrics that emphasize deep phylogenetic relationships and hence reflect early evolutionary processes, against metrics that emphasize shallow relationships and more recent evolutionary events[16].

Species richness and phylogenetic diversity of angiosperms vary greatly across the world[3,17,18]. To understand the evolution of global angiosperm diversity, we must further consider the timing of global climatic changes and the climatic aspects that were affected. Temperature and precipitation, as well as their intra-annual extremes and seasonality, have been considered as major environmental factors determining species diversity and composition in an area at a large spatial scale[15,19,20]. The world became drier since the mid Miocene as it became colder[21]. Thus, cold and dry climates both became barriers for clades originating in humid tropical climates to disperse into regions of extra-tropical climates. Some adaptations to freezing may also represent adaptations to drought because a key part of freezing stress is the lack of liquid water. We may, therefore, expect some of the same variation in phylogenetic relatedness to be accounted for by both temperature- and precipitation-related variables. On the other hand, all land plants are faced to some degree with the need to conserve water, so water-saving adaptations are widespread in the angiosperm tree of life[22]. Low temperatures instead require a whole suite of completely novel adaptations and may be less widespread phylogenetically. Assessing their unique and joint effects on geographic patterns of phylogenetic relatedness can shed light on the mechanisms driving the variation of phylogenetic relatedness.

Global climate cooling during the Cenozoic created strong gradients across latitudes not only in extreme climate (e.g., decreasing the lowest monthly temperature with increasing latitude) but also in climate seasonality (e.g., increasing intra-annual temperature variation with increasing latitude)[11,23,24]. Because the Earth was dominated by tropical or subtropical climates during the Cretaceous and the early Tertiary[25,26], as noted above, temperature seasonality was low, even at high latitudes[23]. During subsequent global climate cooling, the decrease in temperatures, particularly winter temperature, at higher latitudes strengthened the latitudinal gradient of temperature seasonality from the Eocene toward the present. Thus, assessing unique and joint effects of climate extremes (e.g., minimum temperature and precipitation within a year) and climate seasonality (e.g., temperature and precipitation seasonality) on geographic patterns of phylogenetic relatedness can provide insights into the mechanisms driving the variation of phylogenetic relatedness.

In addition to current climate conditions, other major drivers of phylogenetic relatedness of species in a region include historical climatic conditions and environmental heterogeneity within the region[15]. Regions with a relatively stable climate often have many more species persisting in situ and experience lower rates of species extinction and higher rates of speciation compared to regions with unstable climates during glacial-interglacial cycles[27]. Regions of unstable climates are likely to experience frequent local extinction events, reducing the number of old clades[28]. Thus, stable historical climatic conditions have been found to be positively associated with taxonomic and phylogenetic diversity[29,30]. Further, high topographical and environmental heterogeneity often leads to high species diversity because species with different ecological niches can coexist within a region encompassing different environments. Movements of species up or down on an elevational gradient could act as a local buffer against climatic variation, enabling the persistence of clades. Topographical and environmental heterogeneity promotes species persistence within a region during unfavorable times[31], reducing the risk of extinction on the one hand and often creating dispersal barriers that may increase speciation rates on the other hand[32]. Knowledge of the relative importance of each of the three types of drivers (i.e., current climate, historical climate, and environmental heterogeneity) on phylogenetic relatedness of a clade is crucial to understanding the formation and maintenance of patterns of phylogenetic relatedness of clades.

A recent study[18] investigated geographic patterns of phylogenetic structure (including phylogenetic diversity and relatedness) of angiosperms across the world, but it did not include any analysis of the relationships between phylogenetic structure and environmental drivers. Although several studies have addressed such relationships for angiosperms at regional scales[13,15,33], such studies for angiosperms at a global scale remain lacking. Here, we fill this knowledge gap.

In this study, we explore the relationships between different measures of phylogenetic structure and environmental variables for angiosperms in regional floras across the world and for each individual continental (biogeographic) region. In particular, we address the following three questions: (1) Which of the three types of environmental variables noted above (i.e., current climate, historical climate, environmental heterogeneity) is most important in affecting geographic patterns of phylogenetic structure of angiosperms across the globe? (2) Are precipitation-related climatic variables more important drivers of angiosperm phylogenetic structure than temperature-related climatic variables, or vice versa? (3) Are climate extreme variables (e.g., minimum temperature in winter and minimum precipitation in the driest season) more important drivers of angiosperm phylogenetic structure than climate seasonality variables (e.g., temperature and precipitation seasonality), or vice versa?

Considering that angiosperms originated and diversified in warm, moist, and aseasonal climates, as noted above, and thus these climates are the ancestral niche of angiosperms, following the tropical niche

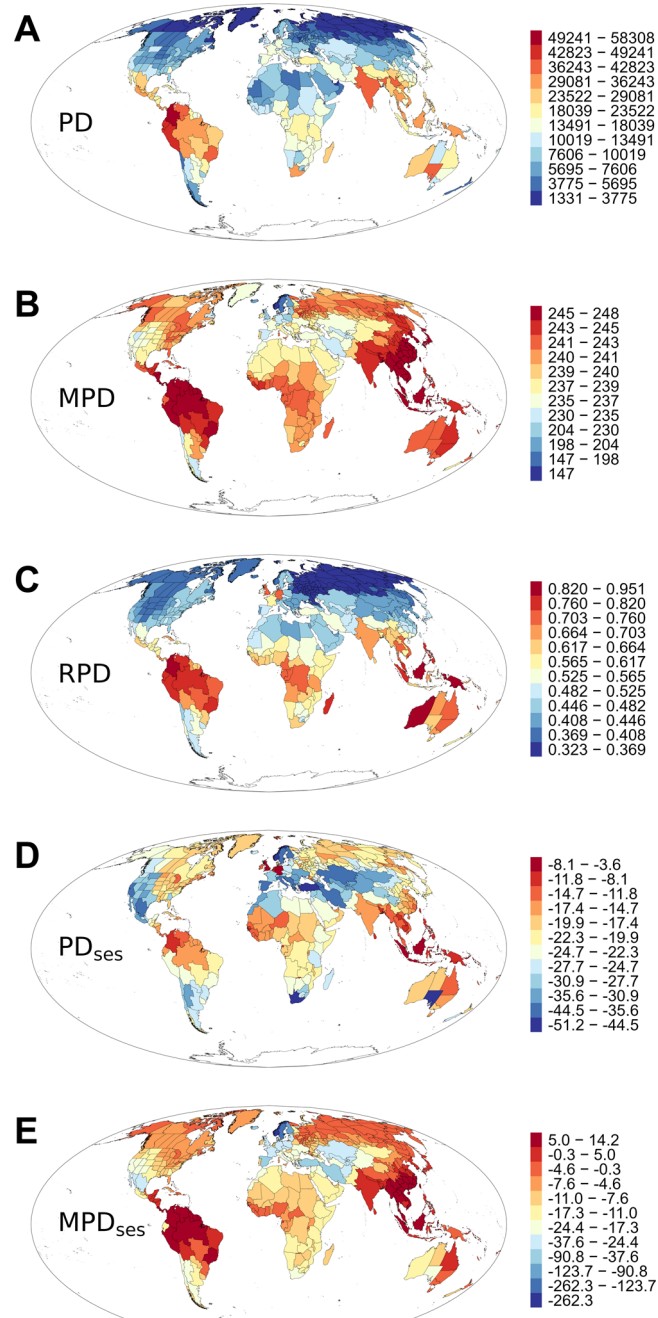

**Fig. 1 | Geographic variation of each of the phylogenetic metrics investigated in this study.** Full names and abbreviations of phylogenetic metrics: **A** phylogenetic diversity (PD), **B** mean pairwise distance (MPD), **C** relative phylogenetic diversity (RPD), **D** the standardized effect size of phylogenetic diversity ($PD_{ses}$), and **E** the standardized effect size of mean pairwise distance ($MPD_{ses}$).

phylogenetic diversity and dispersion with environmental heterogeneity because previous studies on angiosperms at a regional or continental scale (e.g.,[15]) have obtained mixed results on the relationships between metrics of phylogenetic structure and environmental heterogeneity.

## Results

### Relationships between phylogenetic metrics and environmental variables

Phylogenetic diversity and dispersion varied greatly across the world (Fig. 1). The relationship between each phylogenetic metric and each environmental variable, which was measured as the standardized coefficient of regression based on either global or continental models, is shown in Fig. 2. When the relationship was assessed at the global extent and averaged across the five phylogenetic metrics, of the 13 environmental variables, minimum temperature ($T_{min}$) had the strongest effect on the phylogenetic metrics (the average of the absolute values of the standardized coefficients of the five phylogenetic metrics was 0.338). Of the three types of environmental variables, the current climatic variables had the strongest effect, and the historical climatic variables had the weakest effect (the average of the absolute values of the standardized coefficients of the five phylogenetic metrics was 0.250 and 0.121, respectively). When the relationship between each phylogenetic metric and each environmental variable was assessed at the spatial extent of the biogeographic continent, the general patterns of the relationships were similar to those derived from the global models (Fig. 2). Of the five phylogenetic metrics, the average of the absolute values of the standardized coefficients from the 13 models was largest in PD and smallest in $MPD_{ses}$, regardless of whether global or continental models were considered (0.264 versus 0.115 in the former; 0.245 versus 0.170 in the latter).

At the global scale, the 13 environmental variables together explained, on average, 43% of the variation in each of the five phylogenetic metrics, ranging from 23% ($MPD_{ses}$) to 64% (RPD) (Supplementary Table 1). When regional floras of each of the six biogeographic continents were analyzed separately, the average of the variations of the five phylogenetic metrics explained by the 13 environmental variables increased to 70%, ranging from 55% (for Africa) to 87% (for Southern America) (Table S1).

### Variation in phylogenetic metrics explained by different types of environmental variables

When the five phylogenetic metrics were considered collectively, current and historical climatic variables together explained, on average, 34.4% of the variation in the metrics when data at the global scale were analyzed, and 51.2% of the variation in the metrics when data at the continental scale were analyzed, ranging from 33.3% in Africa to 66.3% in Northern America (Fig. 3). The current climatic variables independently explained more variation than did independently the historical climatic variables at the global scale and in five of the six continents (Fig. 3). The current climatic variables independently explained much more variation in the phylogenetic metrics than did jointly by current and historical climatic variables across the globe as a whole and in each of the six continents (Fig. 3).

Current climatic variables and environmental heterogeneity variables together explained, on average, 37.3% of the variation in the metrics when data at the global scale were analyzed and 54.9% of the variation in the metrics when data at the continental scale were analyzed (Fig. 3). The variation in the phylogenetic metrics that was independently explained by the current climatic variables was greater than that independently explained by the environmental heterogeneity variables both at the global scale and in all the six continents (Fig. 3); it was also greater than that jointly explained by current climatic variables and environmental heterogeneity variables at the global scale and in five of the six continents (Fig. 3).

conservative hypothesis[3], we predict that phylogenetic diversity and phylogenetic dispersion, the latter of which is negatively related to phylogenetic relatedness (i.e., greater phylogenetic dispersion reflects lower phylogenetic relatedness and lower phylogenetic clustering), decrease with decreasing temperature and precipitation and with increasing seasonality. Previous studies on angiosperms at a regional or continental scale (e.g.[15]) found that phylogenetic diversity and dispersion are weakly, and often non-significantly, related with measures of the Quaternary climate change, accordingly, we do not make predictions for the relationships of phylogenetic diversity and dispersion with historical climate. Similarly, we do not predict the relationships of

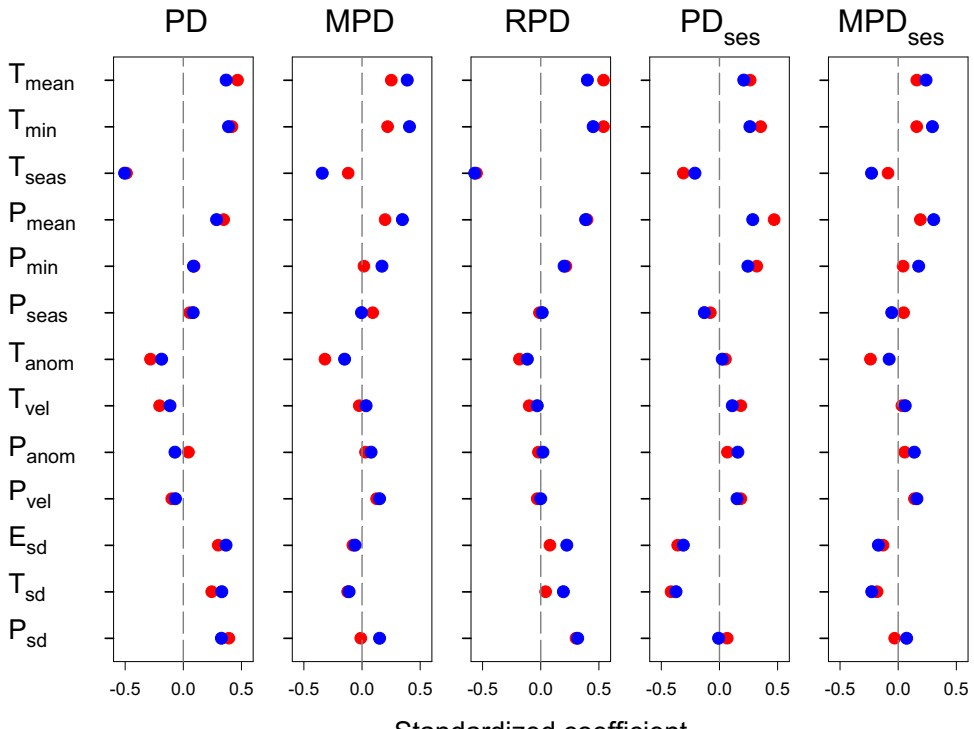

**Fig. 2 | Standardized coefficient of regression between each of phylogenetic metrics and each of climatic and historical climatic variables and within-region variability of elevation, temperature, and precipitation.** A red dot represents a standardized coefficient derived from a single model including all regions across the world; a blue dot represents the average value of six standardized coefficients derived from six models, each of which included regions in one of the six biogeographic continents. Abbreviations of phylogenetic metrics: PD area-corrected phylogenetic diversity, MPD mean pairwise distance, RPD relative phylogenetic diversity, $PD_{ses}$ standardized effect size of phylogenetic diversity, $MPD_{ses}$ standardized effect size of mean pairwise distance; abbreviations of current climatic variables: $T_{mean}$ mean annual temperature, $T_{min}$ minimum temperature of the coldest month, $T_{seas}$ temperature seasonality, $P_{mean}$ annual precipitation, $P_{min}$ precipitation during the driest month, $P_{seas}$ precipitation seasonality; abbreviations for historical climatic variables are: $T_{anom}$ temperature anomaly, $T_{vel}$ temperature velocity, $P_{anom}$ precipitation anomaly, $P_{vel}$ precipitation velocity; abbreviations of within-region variability: $E_{sd}$ standard deviation of elevation, $T_{sd}$ standard deviation of mean annual temperature, $P_{sd}$ standard deviation of annual precipitation. Source data are provided as a Source Data file.

When the phylogenetic metrics were related to historical climatic variables and environmental heterogeneity variables, these two groups of environmental variables collectively explained 20.9% of the variation in the metrics when data at the global scale were analyzed, and 30.7% of the variation in the metrics when data at the continental scale were analyzed (Fig. 3). Environmental heterogeneity variables independently explained more variation than did independently the historical climatic variables at the global scale and in three of the six continents (Fig. 3).

When averaged across the six continental regions for each of the five phylogenetic metrics, the relative power of one type over the other type of environmental variables in each of the three pairs was similar to that reported above (Fig. 4). For example, for each of the phylogenetic metrics, current climatic variables independently explained much more variation in the metric than did either historical climatic variables or environmental heterogeneity variables. On average, across the three pairwise major types of environmental variables, the total explained variation in each phylogenetic metric was 42.1%, 51.8%, 45.3%, 47.5%, and 40.7% for PD, MPD, RPD, $PD_{ses}$, and $MPD_{ses}$, respectively.

When all three major types of explanatory variables were considered collectively, they explained 24.2–68.5% of the variation in the phylogenetic metrics (Fig. 5). More than one-third of the explained variation was explained independently by current climatic variables. The variation in the phylogenetic metrics that was explained jointly by two or three major types of explanatory variables was less than 11% (Fig. 5).

At the global scale, the six current climatic variables together explained, on average, 29.2% of the variation in each phylogenetic

metric, ranging from 7.9% in $MPD_{ses}$ to 60.7% in RPD. The variation explained jointly by temperature- and precipitation-related variables was greater than that explained independently by temperature-related variables or by precipitation-related variables, of which the latter explained more variation than the former (Fig. 6). However, at the continental scale, different continents showed different patterns. For example, temperature-related variables were more important than precipitation-related variables in Europe, Australasia, and Southern America, whereas the opposite pattern was observed in the other three continents (Fig. 6). The variation in each phylogenetic metric that was explained uniquely by climatic extreme variables was, on average, similar to that by climatic seasonality variables, either of which explained less variation, compared to that explained jointly both at the global scale and in each of the six continents (Fig. 6).

When individual phylogenetic metrics were considered separately and data were analyzed for each of the six continents separately, on average, temperature-related variables independently explained more variation than did precipitation-related variables in four of the five phylogenetic metrics (i.e., all but RPD; Fig. 7). When climatic extreme variables were compared with climatic seasonality variables, the former independently explained more variation than did the latter in three of the five phylogenetic metrics (i.e., RPD, $PD_{ses}$, and $MPD_{ses}$; Fig. 7).

## Discussion

In the present study, we investigated the relationships of five metrics of phylogenetic community structure in regional angiosperm floras across the world with 13 environmental variables in three major groups

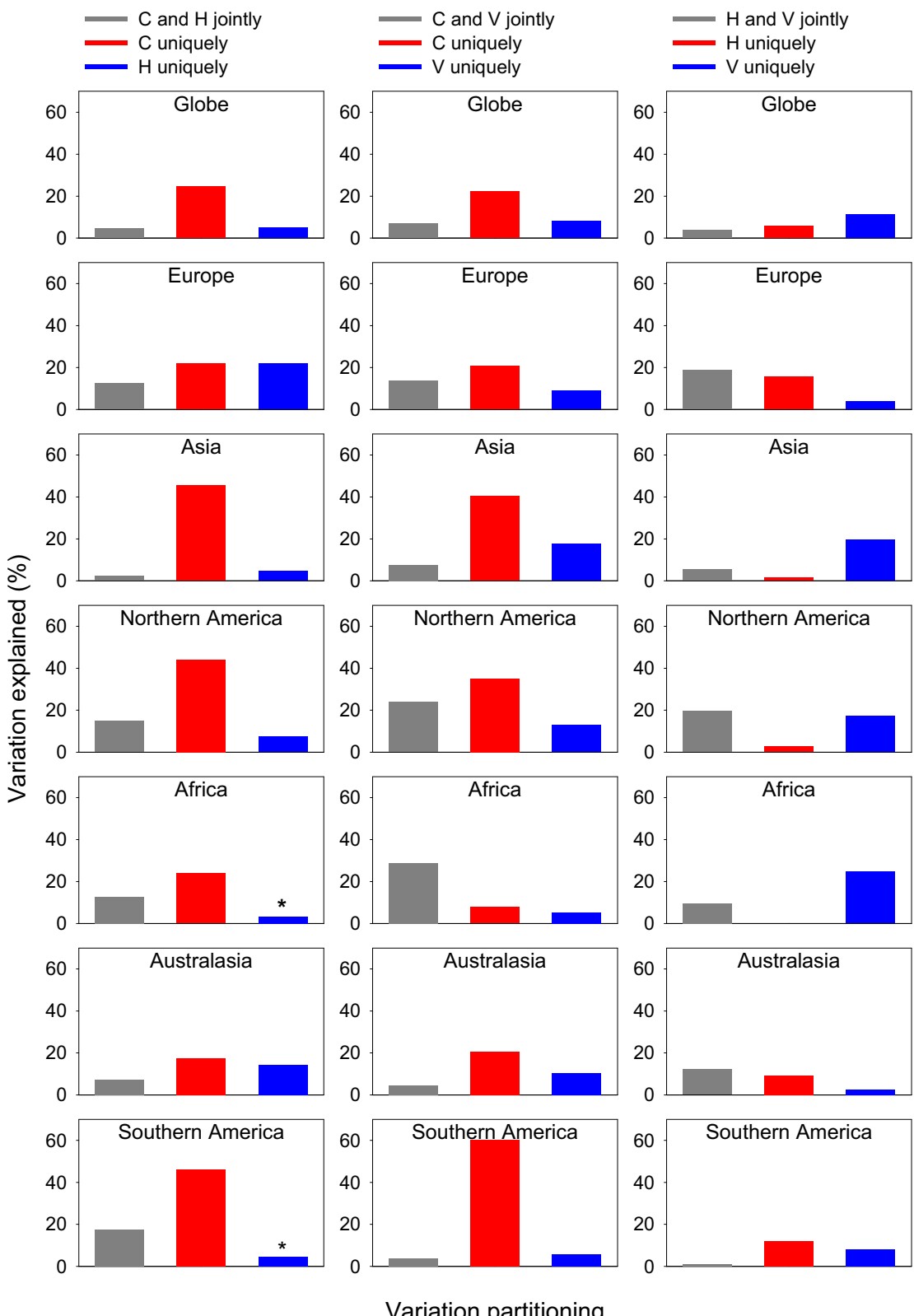

**Fig. 3 | Partition of the variation in phylogenetic metrics of angiosperms in geographic regions across the globe and within biogeographic continents explained by current climatic variables (C), historical climatic variables (H), and within-region variability of elevation, temperature, and precipitation (V).** Each variation partitioning analysis included two sets of explanatory variables. Each bar represents the average value of five phylogenetic metrics (i.e., PD, MPD, RPD, $PD_{ses}$, and $MPD_{ses}$). Note that the total amount of the variation explained by the pairwise groups of explanatory variables in each panel is the sum of the three bars in the panel (values for some bars may be too small to be seen). An asterisk above a bar represents a negative value resulting from the variation partition. See Methods for explanatory variables in each of the three groups. Source data are provided as a Source Data file.

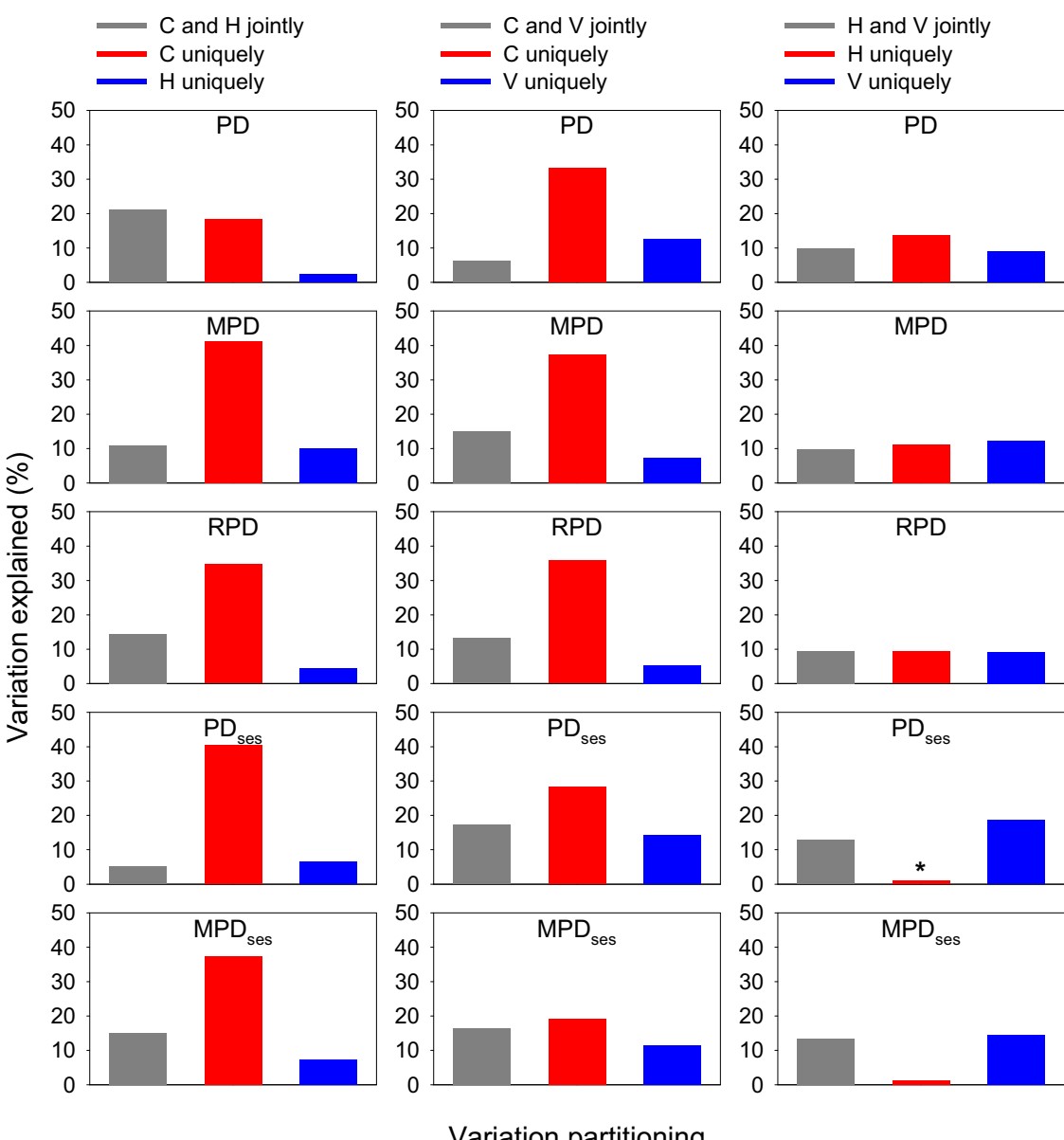

**Fig. 4 | Partition of the variation in phylogenetic metrics of angiosperms in geographic regions within biogeographic continents explained by current climatic variables (C), historical climatic variables (H), and within-region variability of elevation, temperature, and precipitation (V).** Each variation partitioning analysis included two sets of explanatory variables. Each bar represents the average value of the six biogeographic continents. Note that the total amount of the variation explained by the pairwise groups of explanatory variables in each panel is the sum of the three bars in the panel. See Methods for explanatory variables in each of the three groups. Abbreviations of phylogenetic metrics: PD area-corrected phylogenetic diversity, MPD mean pairwise distance, RPD relative phylogenetic diversity, PD$_{ses}$ standardized effect size of phylogenetic diversity, MPD$_{ses}$ standardized effect size of mean pairwise distance. An asterisk above a bar represents a negative value resulting from the variation partition. Source data are provided as a Source Data file.

representing current climate, historical climate, and environmental heterogeneity. The key findings of this study at the global scale include the following. (1) Of the five phylogenetic metrics examined in the study, the geographic variation of relative phylogenetic diversity (RPD) is most strongly associated with the geographic variation of the 13 environmental variables as a whole (Fig. 5). (2) The 13 environmental variables explained more variation in the metrics of phylogenetic structure representing shallow evolutionary history (PD and PD$_{ses}$) compared to those emphasizing deep evolutionary history (MPD and MPD$_{ses}$) (Fig. 5). (3) Current climatic variables explained, in general, more variation in the phylogenetic structure metrics than did historical climatic variables or environmental heterogeneity. (4) The variation of phylogenetic metrics explained jointly by temperature- and precipitation-related variables of current climate was greater than that independently by either type of the variables, and precipitation-related variables explained slightly more variation than did temperature-related variables (Fig. 6). (5) Variables representing climate extremes explained more variation in phylogenetic metrics than did variables representing climate seasonality (Fig. 7). (6) When individual biogeographic continents were considered separately, the above-summarized patterns did not hold for some continents, reflecting the effect of region-specific factors on patterns of phylogenetic structure.

At the global scale, current climate, historical climate, and environmental heterogeneity together explained over two-thirds (69%) of the variation in relative phylogenetic diversity (RPD), which is much larger than the amount of the explained variation in the other phylogenetic metrics (24–54%). RPD quantifies the relative importance of

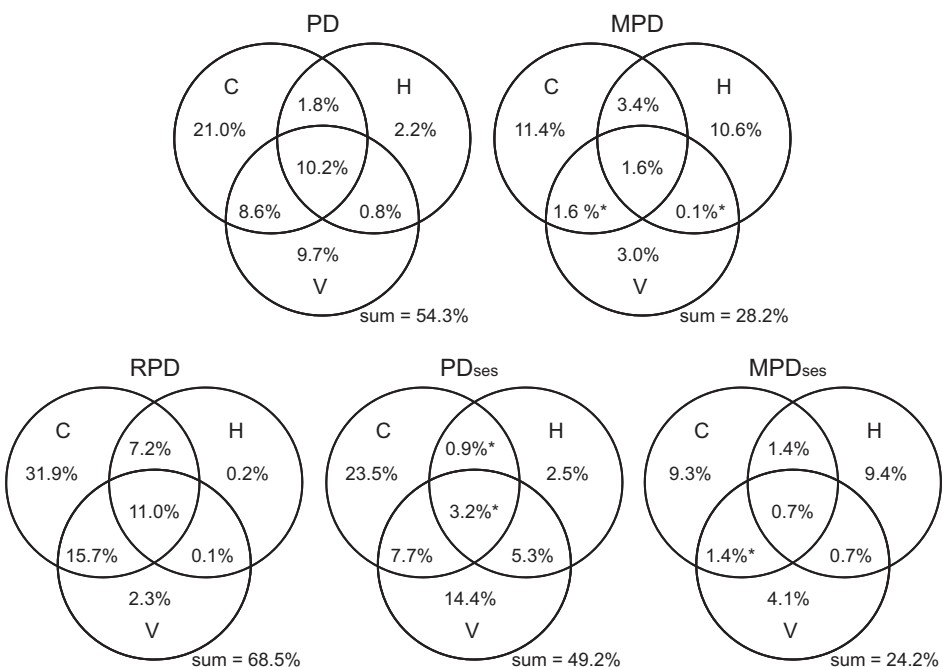

**Fig. 5 | Pure and shared effects of three sets of explanatory variables on phylogenetic metrics of regional angiosperm floras across the world.** Abbreviations of the three sets of explanatory variables: C current climatic variables, H historical climatic variables, V variability of elevation, temperature, and precipitation within the region. See Methods for details about the three sets of explanatory variables. Asterisks indicate negative values resulting from variation partitioning analysis.

phylogenetic overdispersion and clustering that reflect signals of biogeographic history and ecological processes. Regions with high RPD are thought to have experienced high diversification rates of multiple clades or immigration of multiple clades that radiated successfully, whereas regions with low RPD may be associated with large radiations of a few closely related clades[34,35]. High RPD may indicate refugial areas or the effect of competitive exclusion, whereas low RPD may result from recent diversification or habitat filtering[36]. We found high values of RPD mainly in tropical regions across the globe, perhaps reflecting refugial conditions for lineages that, through niche conservatism, have not been able to adapt to temperate conditions. The refugial aspect is emphasized by the extremely high values of RPD (relative to adjacent regions) in regions well known to have acted as biotic refugia, such as Madagascar, Borneo, New Guinea, and Western Australia (e.g.,[36–38]). Conversely, low values of RPD in temperate regions are likely to result from a combination of climatic niche filtering coupled with recent diversification of cold-adapted lineages. Unexpectedly, we found high values of RPD in Germany and the British Isles. We suspect that this may result from different taxonomic concepts applied to these floras than from any real biological difference relative to other temperate regions, but this remains to be tested by comparing different taxonomic arrangements.

Metrics measuring phylogenetic structure reflecting shallow evolutionary history (i.e., PD and PD$_{ses}$; 54% and 49%, respectively) were much better explained by the environmental variables than those measuring phylogenetic structure reflecting deep evolutionary history (i.e., MPD and MPD$_{ses}$; 28% and 24%, respectively) (Fig. 5). These patterns are consistent with those observed in regional studies of angiosperm phylogenetic structure. For example, in regional angiosperm floras in China, PD$_{ses}$ are more strongly correlated with mean annual temperature and annual precipitation than MPD$_{ses}$[15]. We propose four mutually non-exclusive explanations for this result. First, the majority of angiosperm species are phylogenetically young, in many cases resulting from radiations within a few tens of million years (e.g.,[33,39]), so phylogenetic metrics are more strongly influenced by these species. Second, although extinction plays a role at various

levels of the phylogeny, we consider it likely that recent radiations are relatively more strongly influenced by speciation than by extinction, whereas extinctions play a stronger role in older families[40,41]. Since extinction events are only indirectly reflected in a phylogeny based on extant taxa, phylogenetic metrics reflecting deep relationships may be less informative for detecting the effect of extinction. Third, along a similar line of thought, variation in the phylogenetic metrics was best explained by current climate alone or jointly with other environmental variables, and recent past climate is expected to influence recent radiation of clades more strongly than radiations of clades at a deep evolutionary history. Therefore, it makes sense that environmental variables are associated more strongly with phylogenetic metrics reflecting shallow evolutionary history, which were driven largely by recent radiations of clades, than with phylogenetic metrics reflecting deep evolutionary history, which were driven by ancient evolutionary and historical events (e.g., massive extinctions and plate tectonics) more than by recent evolutionary and historical events (e.g., speciation, extinction, and dispersal during glacial–interglacial cycles of the late Cenozoic)[42]. Finally, models of current climate are more accurate than those of past conditions because they can be calibrated with station data, whereas models of past climates are based on very general global assumptions[43]. All of these factors in combination lead to a situation where the effects of recent diversification events are more easily explained than those in the deep past. Interestingly, our finding that environmental variables explain more variation in phylogenetic metrics reflecting shallow evolutionary history than in those reflecting deep evolutionary history is contrary to the finding for regional fern floras across the world[44]. This discrepancy might result from differences in evolutionary histories between the two groups of plants. For example, current geographic patterns of species composition and diversity in many old clades of ferns reflect massive extinctions of these clades in the Mesozoic[45–47]; this is not the case for old clades in angiosperms.

Of the three types of environmental variables (i.e., current climate, historical climate, and environmental heterogeneity), current

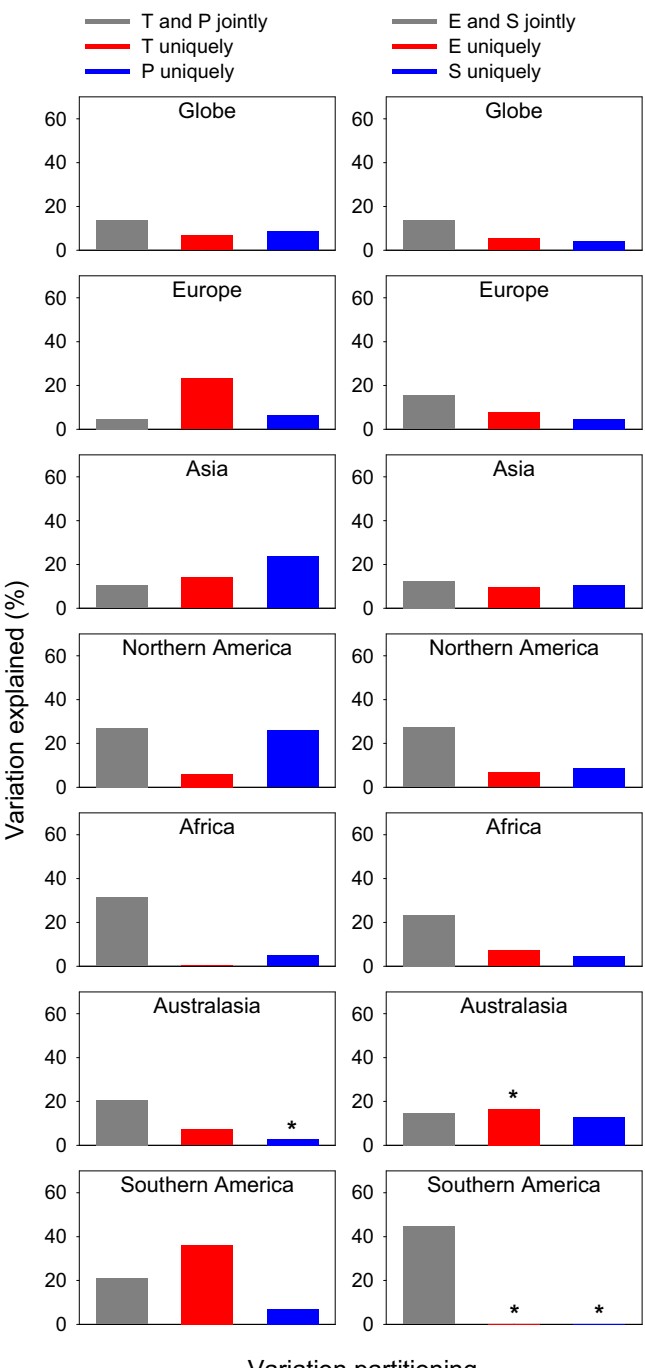

**Fig. 6 | Partition of the variation in phylogenetic metrics of angiosperms in geographic regions across the globe and within biogeographic continents explained by current climatic variables, summarized for the globe or each continent.** Each variation partitioning analysis included two sets of explanatory variables (Left column: temperature-related (*T*) versus precipitation-related (*P*) variables; right column: climate extreme (*E*) versus climate seasonality (*S*)). Each bar represents the average value of five phylogenetic metrics (i.e., PD, MPD, RPD, $PD_{ses}$, and $MPD_{ses}$). Note that the total amount of the variation explained by the pairwise groups of explanatory variables in each panel is the sum of the three bars in the panel (values for some bars may be too small to be seen). An asterisk above a bar represents a negative value resulting from the variation partition. See Methods for explanatory variables in each group of climatic variables. Source data are provided as a Source Data file.

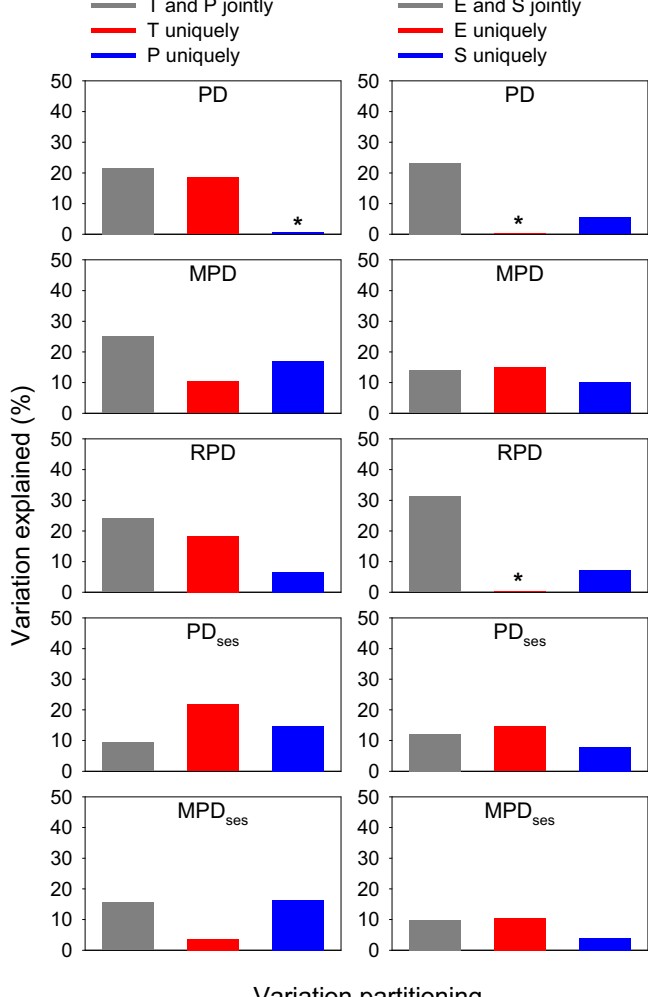

**Fig. 7 | Partition of the variation in phylogenetic metrics of angiosperms in geographic regions within biogeographic continents explained by current climatic variables, summarized for each phylogenetic metric.** Each variation partitioning analysis included two sets of explanatory variables (Left column: temperature-related (*T*) versus precipitation-related (*P*) variables; right column: climate extreme (*E*) versus climate seasonality (*S*)). Each variation partitioning analysis included two sets of explanatory variables. Each bar represents the average value of the six biogeographic continents. Note that the total amount of the variation explained by the pairwise groups of explanatory variables in each panel is the sum of the three bars in the panel (values for some bars may be too small to be seen). An asterisk above a bar represents a negative value resulting from the variation partition. See Methods for explanatory variables in each group. Abbreviations of phylogenetic metrics: PD = area-corrected phylogenetic diversity, MPD mean pairwise distance, RPD relative phylogenetic diversity, $PD_{ses}$ standardized effect size of phylogenetic diversity, $MPD_{ses}$ standardized effect size of mean pairwise distance. Source data are provided as a Source Data file.

climate explains, in general, more variation in the phylogenetic metrics than either historical climate or environmental heterogeneity. In most cases, a large amount of the variation explained by the current climate for a given phylogenetic metric is not overlap with the variation explained jointly by two or three types of environmental variables (Fig. 5). This suggests that current climatic conditions are the main drivers of global patterns of phylogenetic structure of angiosperms in regional floras across the world, although other factors not tested here

might be involved. When the relative effects of historical climate and environmental heterogeneity on phylogenetic structure are compared, interestingly, environmental heterogeneity is a stronger driver of phylogenetic structure reflecting shallow evolutionary history (i.e., PD and PD$_{ses}$) whereas historical climate is a stronger driver of phylogenetic structure reflecting deep evolutionary history (MPD and MPD$_{ses}$) (Fig. 5). We suggest that the relationship of environmental heterogeneity to PD and PD$_{ses}$ may have two reasons, both linked to species diversification and the radiation of clades. First, environmental heterogeneity leads to the spatial separation of species populations, facilitating allopatric speciation. Second, it is linked to niche differentiation between species, favoring radiations involving an environmentally adaptive component.

Focusing on the different current climatic variables examined in this study, we found that temperature- and precipitation-related variables jointly explained more variation in the phylogenetic metrics than either type of the variables explained independently. This has likely resulted from two causes. First, there is some covariance of these factors (e.g., Spearman rank correlation is 0.38 between mean annual temperature and annual precipitation for the geographic regions of this study), so a statistical separation of their effects reflected in the covariance is not possible. Second, and possibly more importantly, although the physiological features required to adapt to either type of stress are different, both cold and drought stress may influence the phylogenetic structure of angiosperm assemblages in similar ways, selecting for few clades that have evolved the necessary adaptations. Looking at the variation independently explained by the two sets of climatic factors, we found that precipitation-related variables explained slightly more variation than did temperature-related variables, but whether this is biologically meaningful or simply an effect of the data structure is unclear, and we deem the differences not worthy of further exploration.

On the other hand, we found that variables representing climate extremes explained more variation in phylogenetic metrics than variables representing climate seasonality. This makes biological sense in that seasonality is a regularly occurring event to which plants can adapt, whereas climatic extremes only occur occasionally, making it more difficult for plants to adapt[48]. For instance, developing resistance to extreme cold is physiologically very costly for angiosperm trees, resulting in lower growth rates and, hence, lower competitive ability[49,50]. In such a situation, the species are faced with a tradeoff for optimizing for fast growth on the one hand and for protecting against the occasional extreme cold on the other hand, so many will not be optimally adapted to extremely low temperatures occurring in their distributional range. Under such circumstances, climatic extremes are likely to act as strong filters of community assembly.

Finally, we found that the variation of each phylogenetic metric that was explained by the environmental variables at the continental scales was higher than that at the global scale (e.g., on average, 70% versus 43%; Table S1). This indicates that continental models better describe the relationship between phylogenetic structure and environment in their respective continents compared to global models. We found that for a given phylogenetic metric, the variation explained by the environmental variables varied greatly among continents (Table S1). For example, averaging across the five phylogenetic metrics and comparing the six continents, the environmental variables explained the most (87%) variation in Southern America and the least (55%) variation in Africa. This is consistent with the findings of Qian et al.[44] for ferns, where the variation in phylogenetic metrics that was explained by environmental variables varied greatly among the continents, and the environmental variables explained the least variation in Africa. However, because phylogenetic metrics and environmental variables examined in the two studies are not identical, a direct comparison between the findings of the two studies cannot be made.

Nevertheless, it would appear that the fern and angiosperm floras in Africa are phylogenetically unique and less in balance with environmental conditions than on other continents, presumably reflecting the massive extinction events that took place there in the Miocene[51]. The great variation in the explained variation of phylogenetic metrics among continents likely reflects that different regional and historical processes in different regions have played an important role in driving patterns of phylogenetic structure. For instance, the Southern American flora was strongly influenced by the rise of the Andes and the associated development of novel ecosystems[52], whereas in the Sundaland and Melanesian regions, island biogeographical processes dominated[53].

These continental differences also emerged when considering different climatic factors. Globally, temperature- and precipitation-related variables affected phylogenetic structure more or less equally. However, when individual continents were considered separately, temperature-related variables played a more important role in Europe, Australasia, and Southern America, whereas precipitation-related variables played a more important role in Asia, Northern America, and Africa. This pattern of the relative effects of these two types of current climatic variables on phylogenetic structure for angiosperms in different continents is largely similar to that for ferns[44]. Similar mixed patterns were also observed when comparing the relative effects of climate extreme and climate seasonality on the phylogenetic structure.

Concluding, we found that the phylogenetic structure of angiosperms globally shows clear and meaningful relationships with environmental factors. Current climatic conditions have the highest predictive power, especially on phylogenetic metrics reflecting recent evolutionary relationships. These are also related to current environmental heterogeneity, presumably because this favors plant speciation in various ways. However, we also found evidence that past climatic conditions, and particularly refugial conditions, play an important role in determining the phylogenetic structure of regional floras. Finally, the relationships between environmental conditions and phylogenetic metrics differed among continents, reflecting the different evolutionary histories of their floras. Considering these results together, we find that the phylogenetic structure of angiosperm floras worldwide is influenced by a combination of factors, starting with the environmental conditions and the geographic setting at the onset of the angiosperm radiation some 100–150 mya, and culminating with recent radiations influenced by current climate and topographical conditions. Beyond these general patterns for the entire angiosperm flora, it is likely that different clades of angiosperms each had their own evolutionary history, resulting in idiosyncratic patterns, and opening fascinating opportunities for further research on the group-specific influences of factors such as physiological adaptations to frost and drought, or the evolution of different life forms or dispersal and pollination modes, all of which will influence the resulting phylogenetic structure. In this sense, our study sets the general stage, but each individual actor likely will tell its own story.

## Methods
### Regional species assemblages
Our geographic sampling units are 391 regions, as shown in Supplementary Fig. 1, which were based on Brummitt[54] and Zhang et al. [55]. Species lists of native angiosperms in these regions were compiled based on Plants of the World Online (http://www.plantsoftheworldonline.org), on which the World Checklist of Vascular Plants (WCVP) is based[56], and World Plants (https://www.worldplants.de), which were supplemented with data from other sources[57,58]. We used the package U.Taxonstand[59] to standardize botanical nomenclature according to World Plants (https://www.worldplants.de) and combined distributions of infraspecific taxa with

those of their respective species. As a result, 341,846 angiosperm species were included in this study.

## Phylogeny

We used the megatree GBOTB.extended.WP.tre[60] as a phylogenetic backbone to generate a phylogeny for the species of this study. The megatree was derived from the megatrees presented in Smith and Brown[39] and Zanne et al. [12]. We used the functions build.nodes.1 and Scenario 3 in the package U.PhyloMaker[61] to add species to the megatree. These methods have been commonly used in generating plant phylogenies for ecological and biogeographic studies (e.g.,[62–67]), and Qian and Jin[68] showed that values of phylogenetic metrics derived from a phylogeny generated with these methods are nearly perfectly correlated with those derived from a phylogeny fully resolved at the species level.

## Phylogenetic metrics

Because different metrics of phylogenetic structure capture different aspects of phylogenetic structure, using multiple metrics of phylogenetic structure that quantify different aspects of phylogenetic structure can help gain a better understanding of phylogenetic structure. We used the following five phylogenetic metrics to quantify the phylogenetic structure of regional angiosperm floras in this study: phylogenetic diversity (PD), mean pairwise distance (MPD), relative phylogenetic diversity (RPD), the standardized effect size of phylogenetic diversity ($PD_{ses}$), and the standardized effect size of mean pairwise distance ($MPD_{ses}$). These metrics reflect different aspects of phylogenetic structure. Specifically, PD is the sum of all phylogenetic branch lengths that connect species in an assemblage[69]. Because PD increases with species richness, which commonly increases linearly with log-transformed sampling area[17], we divided PD in each region by the $log_{10}$-transformed area (in square kilometers) of the region to account for variation in sampling area[15,18,44]. Area-corrected PD in each region was considered as a measure of PD in the region in this study. MPD is the mean phylogenetic pairwise distance (i.e., branch length) among all pairs of species within an assemblage[70], and is mathematically independent of species richness[71]. RPD is PD observed on the original tree divided by PD observed on a comparison tree, with both trees being scaled such that branch lengths are calculated as a fraction of the total tree length[36]. High RPD indicates an over-representation of long branches, whereas low RPD indicates an over-representation of short branches. $PD_{ses}$ and $MPD_{ses}$ measure the phylogenetic dispersion of species assemblages at different evolutionary depths and thus represent the legacy of evolutionary histories at different phylogenetic depths: $MPD_{ses}$ measures the more basal structure of the phylogenetic tree, whereas $PD_{ses}$ measures the more terminal structure of the phylogenetic tree[15,16]. A positive value of $PD_{ses}$ or $MPD_{ses}$ reflects relative phylogenetic overdispersion of species, while a negative value reflects relative phylogenetic clustering of species. We used the package PhyloMeasures[72] to calculate PD, MPD, $PD_{ses}$, and $MPD_{ses}$, and the package canaper[73] to calculate RPD. We used a Mollweide (equal-area) projection to map each of the five phylogenetic metrics (Fig. 1).

## Climate data

We related the aforementioned phylogenetic metrics to three sets of explanatory variables: a set of variables reflecting current climate conditions, a set of variables reflecting historical climate change since the Last Glacial Maximum, and a set of variables reflecting niche heterogeneity. The set of current climatic variables included mean annual temperature ($T_{mean}$), minimum temperature of the coldest month ($T_{min}$), temperature seasonality ($T_{seas}$), annual precipitation ($P_{mean}$), precipitation during the driest month ($P_{min}$), and precipitation seasonality ($P_{seas}$). These climatic variables are widely

considered as drivers of plant and animal distributions and biodiversity patterns (e.g.[15,74,75]). The set of historical climatic variables included the differences in mean annual temperature and annual precipitation between the Last Glacial Maximum and the present as a temperature anomaly ($T_{anom}$) and precipitation anomaly ($P_{anom}$), respectively, and temperature velocity ($T_{vel}$) and precipitation velocity ($P_{vel}$) as the ratio of the rate of climate change through time to the rate of climate change across space[30]. Many of angiosperm radiations are deeper in time than the Pleistocene, but previous studies have shown that the association between climate change since an earlier time (e.g., Pliocene) and current plant distributions is much weaker, or none existence, compared with climate change since the Last Glacial Maximum[76], suggesting that climate change since the Pleistocene might be a more important driver of current plant distributions, compared to climate change since an earlier geological time. Thus, we used climate change since the Last Glacial Maximum as a measure of historical climate. The set of variables reflecting environmental heterogeneity (i.e., variability of topography and climate within each region) included the standard deviations of elevation ($E_{sd}$), mean annual temperature ($T_{sd}$), and annual precipitation ($P_{sd}$) within regions at the 30-arc-second resolution. Data for the current climatic variables were obtained from the CHELSA climate database (https://chelsa-climate.org/bioclim)[77] and data on historical climate change were obtained from Sandel et al. [30]. We used a Mollweide (equal-area) projection to map each of the 13 environmental variables (Supplementary Fig. 2).

## Data analysis

We assessed the relationships between phylogenetic metrics and environmental variables using simultaneous autoregressive error models[78]. Specifically, we regressed each of the five phylogenetic metrics on each of the 13 environmental variables and assessed the relationship between pairwise variables based on standardized coefficients of regressions. To determine the effects of different groups of environmental variables independently and jointly on each phylogenetic metric, we conducted multiple sets of variation partitioning analyses[79] to partition the explained variation into multiple portions based on an adjusted coefficient of determination. First, we conducted variation partitioning analyses to determine the relative effects of current climatic variables and historical climate change variables on phylogenetic metrics, which partitioned the amount of the explained variation in a phylogenetic metric into three portions: variation explained uniquely by current climatic variables, variation explained uniquely by historical climate change variables, and variation explained jointly by the two sets of variables. Similarly, we conducted variation partitioning analyses for current climatic variables versus variability of topography and climate and for historical climate change variables versus variables of topographic and climatic variability. Second, we conducted variation partitioning analyses, which each included all three groups of the explanatory variables and partitioned the explained variation into three independent effects and four shared effects[79]. Third, for the current climatic variables, we conducted two variation partitioning analyses for each phylogenetic metric. One analysis determined whether temperature-related variables ($T_{mean}$, $T_{min}$, and $T_{seas}$) or precipitation-related variables ($P_{mean}$, $P_{min}$, and $P_{seas}$) have a stronger influence on the phylogenetic metric, and the other analysis determined whether climate extreme variables ($T_{min}$ and $P_{min}$) or climate seasonality variables ($T_{seas}$ and $P_{seas}$) have a stronger influence on the phylogenetic metric.

In addition to conducting the above-described analyses for the globe as a whole, we also conducted the analyses for each of the six biogeographic continents (as shown in Supplementary Fig. 1; also see Supplementary Note 1). We used the package Spatial Analysis in Macroecology (v4.0; www.ecoevol.ufg.br/sam/; Rangel et al.[80]) for statistical analyses.

## Reporting summary

Further information on research design is available in the Nature Portfolio Reporting Summary linked to this article.

## Data availability

All data needed to evaluate the conclusions in the paper are present in the paper and/or the supplementary materials. Plant distribution data are available in World Plants (WP; https://www.worldplants.de) and Plants of the World Online (POWO; http://www.plantsoftheworldonline.org). Current climate data are available at the CHELSA climate database (https://chelsa-climate.org/bioclim). Data on historical climate change are available at https://doi.org/10.5061/dryad.b13j1[81]. Data used in the analyses reported in this article are provided in Supplementary Data 1. Source data are provided in this paper.

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

## Acknowledgements

M.K. acknowledges support from the Swiss National Science Foundation (grant SNF 310030_188498). J.Z. acknowledges support from the Innovation Program of the Shanghai Municipal Education Commission (grant 2023ZKZD36).

## Author contributions

H.Q. conceived the study, compiled data, conducted analyses, and wrote the first draft of the paper; S.Q. compiled and analyzed data; J.Z. compiled data; M.K. contributed to writing and editing the article. All authors contributed to revising the article.

## Competing interests

The authors declare no competing interests.
