## [Peer Review File · Nature Communications]

Effects of climate and environmental heterogeneity on the phylogenetic structure of regional angiosperm floras worldwideREVIEWER COMMENTS

Reviewer #1 (Remarks to the Author):

The study assessed the relative effects of three groups of environmental predictors (i.e., current climate, historical climate and environmental heterogeneity) on the phylogenetic structure of angiosperm plants at the global scale. Using five different phylogenetic metrics, the authors tested the varied effects of three environmental groups on phylogenetic structure at different evolutionary depths (i.e. shallow vs. deep depths). The study found that current climate is the main drivers of phylogenetic structure consistently. Also, the study showed that the effects of environmental predictors varied across the continents and highlight the importance of regional and historical processes on phylogenetic structure.

However, it is not clear to me just how much of an advance is presented here. Specifically, the importance of current climate to phylogenetic structure of plants has been founded in the previous studies. For example, Qian et al 2019 (PNAS) found that current climate strongly affected phylogenetic structure of seed plants using the same metrics used here (MPDs and PDs). Similarly, Cai et al 2023 (New Phytologist) found that global patterns of phylogenetic diversity for vascular plants is mainly driven by current climate and also highlight the roles of biogeographic history or regional idiosyncrasies.

It is correct that five metrics used in the study capture different aspect of phylogenetic structure. However, it is not clear to me why the authors choose these metrics. For example, relative phylogenetic diversity quantifies to what extent long branches or short branches is present, while PDs and MPDs capture phylogenetic dispersal. These metrics tell totally different story. Is there any reason (e.g., the intrinsic links between these five metrics) so that the authors compared them together?

Importantly, the hypotheses for the relationships between each metric and the environmental variables are missing, taking into account the different aspects captured by these metrics. In the introduction, there are hypotheses about the relationships between environmental conditions and phylogenetic clustering and phylogenetic diversity. However, as important results, the hypotheses about how and why the effects of environmental predictors could vary across different evolutionary depths is still missing, only roughly mentioned.

Climate variables tend to be highly correlated. In particular, both annual means and extremes were included in the study. Have the authors already tested for collinearity between a set of variables? If not, I would appreciate at least one test for this issue. If there is high collinearity, the inclusion of all these variables should be problematic for modelling.

The results in Lines 340-342 and Lines 349-352 are contradictory. Two places both compared the relative importance of current climate and historical climate but with different results. This might be written by mistakes. Please check carefully.

Reference:

Cai, L., Kreft, H., Taylor, A., Denelle, P., Schrader, J., Essl, F., van Kleunen, M., Pergl, J., Pysek, P., Stein, A., Winter, M., Barcelona, J.F., Fuentes, N., Inderjit, Karger, D.N., Kartesz, J., Kuprijanov, A., Nishino, M., Nickrent, D., Nowak, A., Patzelt, A., Pelsner, P.B., Singh, P.,

Wieringa, J.J., & Weigelt, P. (2023a) Global models and predictions of plant diversity based on advanced machine learning techniques. *New Phytologist*, 237, 1432-1445.

Qian, H., Deng, T., Jin, Y., Mao, L., Zhao, D., & Ricklefs, R.E. (2019) Phylogenetic dispersion and diversity in regional assemblages of seed plants in China. *Proceedings of the National Academy of Sciences of the United States of America*, 116, 23192-23201.

Reviewer #2 (Remarks to the Author):

This manuscript describes an investigation into patterns from several phylogenetic diversity metrics and their potential links with environmental variables, both on a global and continental scale. Generally, I have little to say against the work presented here, the approaches they used to address these questions and the conclusions they draw from them. I have a number of mostly minor comments listed below.

In several instances, additional references supporting certain statements would be required, e.g. L80-82, L118-199 (several studies have shown this), and 132-133, to mentioned a few.

L265: You use climate change since the LGM as a measure of historical climates, but you then associate these in the discussion (e.g. L 502-504) to deep time phylogenetic relationships and MPD metrics, which assess patterns at much deeper time scales than the time since the LGM in the present case. It would be essential not to mix these up and avoid sweeping conclusions on the basis of these comparisons.

L461: I wouldn't go as far as saying that metrics predominantly driven by deep phylogenetic relationships are less informative. In the present context, i.e. considering environmental variables, it may be the case, but overall they represent different processes, and are not necessarily less informative.

L461-471: Very long sentence; consider splitting it.

L490: It would be important to state here that among the factors you examined, current climates are the main drivers, but that other factors not tested here might be involved. This is somewhat touched upon in L549-552, but it would need to be more explicit in places.

L538: I am not sure the values of the five phylogenetic metrics used in this study can be averaged as they are not representing the same concept or have the same unit. How did you do this? Is it explained elsewhere?

Fig. 1: I would suggest that you focus this figure on the patterns retrieved for the five phylogenetic diversity metrics and moved the environmental data to the supplemental material. The five metrics are the results and should take prominence here and we can barely see them.

Fig 6: Why use an asterisk to represent negative values rather than extending the graph below zero? There might be an explanation for this, but I haven't looked up Legendre and Legendre 2012 as suggested by the authors (and most readers not familiar with this will probably not either).

Fig 7: I think the start of this legend isn't correct; it seems copied from Fig 6; please check.

Fig S1: You include India and several countries in South-East Asia as part of Australasia. I don't think this is the conventional way to delimit this region and I wonder if this would affect the results, especially the continental comparisons.

REVIEWER COMMENTS

Reviewer #1 (Remarks to the Author):

The study assessed the relative effects of three groups of environmental predictors (i.e., current climate, historical climate and environmental heterogeneity) on the phylogenetic structure of angiosperm plants at the global scale. Using five different phylogenetic metrics, the authors tested the varied effects of three environmental groups on phylogenetic structure at different evolutionary depths (i.e. shallow vs. deep depths). The study found that current climate is the main drivers of phylogenetic structure consistently. Also, the study showed that the effects of environmental predictors varied across the continents and highlight the importance of regional and historical processes on phylogenetic structure.

RESPONSE: Thank you for your effort reviewing our manuscript. We have carefully considered each of your comments. Please see below for our point-by-point responses to your comments.

However, it is not clear to me just how much of an advance is presented here. Specifically, the importance of current climate to phylogenetic structure of plants has been founded in the previous studies. For example, Qian et al 2019 (PNAS) found that current climate strongly affected phylogenetic structure of seed plants using the same metrics used here (MPDses and PDses). Similarly, Cai et al 2023 (New Phytologist) found that global patterns of phylogenetic diversity for vascular plants is mainly driven by current climate and also highlight the roles of biogeographic history or regional idiosyncrasies.

RESPONSE: Yes, some previous studies, including those cited by you, have shown the importance of current climate over historical climate to phylogenetic structure of plants. However, other studies have shown that historical climate is more important than current climate to phylogenetic structure of plants (e.g., Xu et al. 2023. Science Advances, 9:eadd8553). The relative effects of current vs. historical climate, and of environmental heterogeneity remain poorly understood. To our knowledge, our study is the first to assess the relative importance of the three major types of environmental variables (i.e., current climate, historical climate, and environmental heterogeneity) to a variety of phylogenetic metrics for angiosperms at a global scale.

It is correct that five metrics used in the study capture different aspect of phylogenetic structure. However, it is not clear to me why the authors choose these metrics. For example, relative phylogenetic diversity quantifies to what extent long branches or short branches is present, while PDses and MPDses capture phylogenetic dispersal. These metrics tell totally different story. Is there any reason (e.g., the intrinsic links between these five metrics) so that the

authors compared them together?

RESPONSE: Because different metrics of phylogenetic structure capture different aspects of phylogenetic structure, using multiple metrics of phylogenetic structure that quantify different aspects of phylogenetic structure can help gain a better understanding of phylogenetic structure. Using multiple metrics of phylogenetic structure in a study is common (e.g., *Nature Communications* (2014) 5:4473; *Nature* (2018) 554:234-238; *Ecology Letters* (2019) 22:1126–1135; *New Phytologist* (2023) 239:415–428). There are several publications discussing the intrinsic links between different phylogenetic metrics, including those metrics used in our study. Here are some examples:

Cadotte, M.W. & Davies, T.J. (2016) *Phylogenies in Ecology: A Guide to Concepts and Methods* Princeton University Press, Princeton and Oxford.

Mazel, F., Davies, T.J., Gallien, L., Renaud, J., Groussin, M., Münkemüller, T., & Thuiller, W. (2016) Influence of tree shape and evolutionary time-scale on phylogenetic diversity metrics. *Ecography*, 39, 913-920.

Mishler, B.D., Knerr, N., González-Orozco, C.E., Thornhill, A.H., Laffan, S.W., & Miller, J.T. (2014) Phylogenetic measures of biodiversity and neo- and paleo-endemism in Australian *Acacia*. *Nature Communications*, 5, 4473.

Tucker, C., Cadotte, M., Carvalho, S., Davies, J., Ferrier, S., Fritz, S., Grenyer, R., Helmus, M., Jin, L., Mooers, A., Pavoine, S., Purschke, O., Redding, D., Rosauer, D., Winter, M., & Mazel, F. (2016) A guide to phylogenetic metrics for conservation, community ecology and macroecology. *Biological Reviews*, 92, 698-715.

Importantly, the hypotheses for the relationships between each metric and the environmental variables are missing, taking into account the different aspects captured by these metrics. In the introduction, there are hypotheses about the relationships between environmental conditions and phylogenetic clustering and phylogenetic diversity. However, as important results, the hypotheses about how and why the effects of environmental predictors could vary across different evolutionary depths is still missing, only roughly mentioned.

RESPONSE: Because the five phylogenetic structure metrics used in our study are often positively correlated with one another, as shown in the literature, we made some general hypotheses between phylogenetic structure metrics and environmental variables in the last paragraph of the introduction, rather than for each phylogenetic structure metric. And because how and why the effects of environmental predictors could vary across different evolutionary depths remains poorly known, we are not able to make specific hypotheses for them.

Climate variables tend to be highly correlated. In particular, both annual means and extremes were included in the study. Have the authors already tested for collinearity between a set of variables? If not, I would appreciate at least one test for this issue. If there is high collinearity, the inclusion of all these variables

should be problematic for modelling.

RESPONSE: Whether it is appropriate to include highly correlated climate variables in a statistic model depends on what results of the model are to be used. In our study, we used the coefficient of determination of each model (i.e., R-square) to assess the relative importance of each group of explanatory variables via variation partitioning approach. R-square of a model is independent of correlation among explanatory variables. This is easy to be tested. For example, in the case that R-square for the regression of $y = x_1 + x_2$ is 0.7 and there is a high correlation between x_1 and x_2 , when x_1 and x_2 are transformed to two orthogonal, and thus completely unrelated, principal components (i.e., PC1 and PC2) via principal components analysis, R-square for the regression of $y = PC1 + PC2$ remains 0.7. Thus, collinearity among climate variables has no effect on the results of our study.

The results in Lines 340-342 and Lines 349-352 are contradictory. Two places both compared the relative importance of current climate and historical climate but with different results. This might be written by mistakes. Please check carefully.

RESPONSE: There was indeed a writing mistake in Lines 349-352, which has been corrected. Thanks for pointing this out.

Reference:

Cai, L., Kreft, H., Taylor, A., Denelle, P., Schrader, J., Essl, F., van Kleunen, M., Pergl, J., Pysek, P., Stein, A., Winter, M., Barcelona, J.F., Fuentes, N., Inderjit, Karger, D.N., Kartesz, J., Kuprijanov, A., Nishino, M., Nickrent, D., Nowak, A., Patzelt, A., Pelsner, P.B., Singh, P., Wieringa, J.J., & Weigelt, P. (2023a) Global models and predictions of plant diversity based on advanced machine learning techniques. *New Phytologist*, 237, 1432-1445.

Qian, H., Deng, T., Jin, Y., Mao, L., Zhao, D., & Ricklefs, R.E. (2019) Phylogenetic dispersion and diversity in regional assemblages of seed plants in China. *Proceedings of the National Academy of Sciences of the United States of America*, 116, 23192-23201.

Reviewer #2 (Remarks to the Author):

This manuscript describes an investigation into patterns from several phylogenetic diversity metrics and their potential links with environmental variables, both on a global and continental scale. Generally, I have little to say against the work presented here, the approaches they used to address these questions and the conclusions they draw from them. I have a number of mostly

minor comments listed below.

RESPONSE: Thank you for your effort reviewing our manuscript. We have carefully considered each of your comments. Please see below for our point-by-point responses to your comments.

In several instances, additional references supporting certain statements would be required, e.g. L80-82, L118-199 (several studies have shown this), and 132-133, to mentioned a few.

RESPONSE: Thanks for the suggestion. We have cited references in each of the three places.

L265: You use climate change since the LGM as a measure of historical climates, but you then associate these in the discussion (e.g. L 502-504) to deep time phylogenetic relationships and MPD metrics, which assess patterns at much deeper time scales than the time since the LGM in the present case. It would be essential not to mix these up and avoid sweeping conclusions on the basis of these comparisons.

RESPONSE: Thanks for the comment. We have deleted the sentence in Lines 502-504.

L461: I wouldn't go as far as saying that metrics predominantly driven by deep phylogenetic relationships are less informative. In the present context, i.e. considering environmental variables, it may be the case, but overall they represent different processes, and are not necessarily less informative.

RESPONSE: We have modified the concerned text by changing "less informative" to "less informative for detecting the effect of extinction".

L461-471: Very long sentence; consider splitting it.

RESPONSE: We have split the sentence into two.

L490: It would be important to state here that among the factors you examined, current climates are the main drivers, but that other factors not tested here might be involved. This is somewhat touched upon in L549-552, but it would need to be more explicit in places.

RESPONSE: We have included the following text in the line: "although other factors not tested here might be involved".

L538: I am not sure the values of the five phylogenetic metrics used in this study can be averaged as they are not representing the same concept or have the

same unit. How did you do this? Is it explained elsewhere?

RESPONSE: We did not average the values of the five phylogenetic metrics used in this study. Instead, we averaged the values of R-square of regression models derived from each of the five phylogenetic metrics separately. The values of R-square are in the same unit (i.e., varying from 0 to 1, or from 0% to 100%). The same approach of averaging the values of R-square for different phylogenetic metrics has been used in previous studies (e.g., *New Phytologist* (2023) 239: 415–428).

Fig. 1: I would suggest that you focus this figure on the patterns retrieved for the five phylogenetic diversity metrics and moved the environmental data to the supplemental material. The five metrics are the results and should take prominence here and we can barely see them.

RESPONSE: The purpose of Fig. 1 is to show: (1) how the values of a particular phylogenetic metric or environmental variable vary geographically across the world, and (2) how geographic patterns are similar or dissimilar among different variables, particularly between phylogenetic metrics and environmental variables. If we separate the maps of phylogenetic metrics from those of environmental variables and put the maps of the latter in the supplemental material, as you suggested, we are afraid that this might create a burden to readers to compare geographic patterns of phylogenetic metrics with those of environmental variables. Thus, we prefer to retain Fig. 1 as it was. However, if both you and the editor would like us to split Fig. 1 into two, we will do so.

Fig 6: Why use an asterisk to represent negative values rather than extending the graph below zero? There might be an explanation for this, but I haven't looked up Legendre and Legendre 2012 as suggested by the authors (and most readers not familiar with this will probably not either).

RESPONSE: The reason why we used an asterisk to represent negative values, rather than extending the graph below zero, is that some negative values are too small to be seen (e.g., the middle bar of PD in Fig. 7). The way of using an asterisk to represent a negative value has been also used in previous studies (e.g., fig. 4 in *New Phytologist* (2023) 239: 415–428; fig. 3 in *Global Ecology and Biogeography* (2021) 30:1835–1846). We have improved Fig. 6 by enlarging the asterisks in the figure, which can be easily seen.

Fig 7: I think the start of this legend isn't correct; it seems copied from Fig 6; please check.

RESPONSE: Thanks for pointing this out. The legends of the two figures have been differentiated.

Fig S1: You include India and several countries in South-East Asia as part of

Australasia. I don't think this is the conventional way to delimit this region and I wonder if this would affect the results, especially the continental comparisons.

RESPONSE: We included India and several countries in South-East Asia as part of Australasia in order to maximize comparability in geographic extent among continental regions. We conducted an analysis to assess whether including India and several countries in South-East as part of Asia or as part of Australasia will significantly change our result. Specifically, we removed the geographic units of the Indian subcontinents and Indochina from Australasia, and included them in Asia. We then calculated adjusted the R-square values for the relationship between each of the five phylogenetic metrics and each of the three types of environmental variables (i.e., current climate, historical climate, and environmental heterogeneity) in each of the two continental regions. We determined the degree to which the relative importance of two types of environmental variables in each pair differs between our results reported in Figure 3 and those of the new analyses outlined above. We found that the relative importance of each type of environmental variables in each pairwise comparison (e.g., current climate versus historical climate) did not change in 87% of the 30 pairwise comparisons (i.e., 5 metrics by 3 environmental types by 2 continents). Thus, whether including India and several countries in South-East as part of Asia or as part of Australasia did not make a significant difference in our results.

REVIEWERS' COMMENTS

Reviewer #1 (Remarks to the Author):

The authors have addressed most of my previous concerns and comments raised. However, I still have my doubts about the novelty of this study. As I mentioned earlier, the main results (i.e., the importance of current climate for patterns of phylogenetic structure for plants) found in the study have already been shown in previous studies (Qian et al 2019; Cai et al 2023).

Reviewer #2 (Remarks to the Author):

Generally, I think that the authors have addressed my comments and those of the other reviewer. Regarding my comment on Fig.1, I still think that the five metrics should take prominence and the environmental move elsewhere, but I guess it will depend on the final size of this figure. I leave this decision to the Editor. About Fig.6, I still don't understand why the negative values are not shown as below 0 rather than with an asterisk (and I can't see how citing a book would help understand this), but if the Editor is agreeable to this option, it's fine with me. It is great that the authors have tested how the inclusion or not of India in Australasia affects the results. I would suggest that these results are also included here as supplementary data.

Response to the reviewers' comments

REVIEWERS' COMMENTS

Reviewer #1 (Remarks to the Author):

The authors have addressed most of my previous concerns and comments raised.

However, I still have my doubts about the novelty of this study. As I mentioned earlier, the main results (i.e., the importance of current climate for patterns of phylogenetic structure for plants) found in the study have already been shown in previous studies (Qian et al 2019; Cai et al 2023).

RESPONSE: Thank you for your effort reviewing our manuscript. As we responded in the previous round of the review process and show in the manuscript, the main results found in this study differ substantially from those of previous studies.

Reviewer #2 (Remarks to the Author):

Generally, I think that the authors have addressed my comments and those of the other reviewer. Regarding my comment on Fig.1, I still think that the five metrics should take prominence and the environmental move elsewhere, but I guess it will depend on the final size of this figure. I leave this decision to the Editor. About Fig.6, I still don't understand why the negative values are not shown as below 0 rather than with an asterisk (and I can't see how citing a book would help understand this), but if the Editor is agreeable to this option, it's fine with me. It is great that the authors have tested how the inclusion or not of India in Australasia affects the results. I would suggest that these results are also included here as supplementary data.

RESPONSE: Thank you for your effort reviewing our manuscript. Following your suggestion, the revised version of Figure 1 includes only the five phylogenetic metrics, the environmental variables were moved to the supplementary information (Figure S1). With regard to Figure 6, we used an asterisk to indicate a negative value resulting from variation partitioning in order to make the presentation of the results of variation partitioning to be consistent with previous studies, e.g.

fig. 4 in *New Phytologist* (2023) 239: 415–428, doi: 10.1111/nph.18920;

fig. 2 in *Journal of Biogeography* (2023) 50:1817–1825, DOI: 10.1111/jbi.14691;

fig. 5 in *Ecography* (2023) 2023: e06775; doi: 10.1111/ecog.06775;

fig. 2 in *Ecography* (2023) 2023: e06516; doi: 10.1111/ecog.06516

fig. 4 in *Journal of Biogeography* (2022) 49:1911–1919; DOI: 10.1111/jbi.14477;

fig. 3 in *Global Ecology and Biogeography* (2021) 30:1835–1846; DOI:

10.1111/geb.13349

The results of testing whether the inclusion or not of India in Australasia would affect the results of this study have been included in the supplementary information file (Data S1).